# DIFFUSING POLICIES : TOWARDS WASSERSTEIN POLICY GRADIENT FLOWS

## ABSTRACT

Policy gradients methods often achieve better performance when the change in policy is limited to a small Kullback-Leibler divergence. We derive policy gradients where the change in policy is limited to a small Wasserstein distance (or trust region). This is done in the discrete and continuous multi-armed bandit settings with entropy regularisation. We show that in the small steps limit with respect to the Wasserstein distance $W_2$, policy dynamics are governed by the heat equation, following the Jordan-Kinderlehrer-Otto result. This means that policies undergo diffusion and advection, concentrating near actions with high reward. This helps elucidate the nature of convergence in the probability matching setup, and provides justification for empirical practices such as Gaussian policy priors and additive gradient noise.

## 1 INTRODUCTION AND SETTING

Deep reinforcement learning algorithms have enjoyed tremendous practical success at scale Mnih et al. (2015; 2016). Separately, theoretical and practical success through smoothing has also been achieved by generative adversarial networks with the introduction of Wasserstein GANs Arjovsky et al. (2017). In both instances, a smooth relaxation of the original problem has been key to further theoretical understanding. In this work, we take the view of policy gradients iteration through the lens of converging towards a function of the rewards field $r(s, a)$ for a given state $s$. This view uses optimal transport metrized by the second Wasserstein distance rather than the standard Kullback-Leibler divergence. Simultaneously, gradient flows relax and generalize to continuous time the notion of gradient steps. An important mathematical result due to Jordan et al. (1998) shows that in that setting, continuous control policy transport is smooth; this achieved by the heat flow following the Fokker-Planck equation, which also admits a stochastic diffusion representation, and sheds light on qualitative convergence towards the optimal policy. This is to our knowledge the first time that the connection between *variational* optimal transport and reinforcement learning is made.

Policy gradient methods Williams & Peng. (1991); Mnih et al. (2016) look to directly maximize the functional of expected reward under a certain policy $\pi$. $\pi(a|s)$ is the probability of taking action $a$ in state $s$ under policy $\pi$. A policy can hence be identified to a probability measure $\pi \in \mathbb{P}$, the space of all policies. In what follows, functionals are applications from $\mathbb{P} \to \mathbb{R}$. Out of a desire for simplification, we focus on formal derivations, and skip over regularity and integrability questions.

We investigate policy gradients with entropy regularisation in the following setting:

- Bandits, or reinforcement learning with 1-step returns
- Continuous action space
- Deterministic rewards

It is already known Sabes & Jordan. (1996) that entropic regularization of policy gradients leads to a limit energy-based policy that probabilistically matches the rewards distribution. We investigate the dynamics of that convergence. Our contributions are as follows:

1. We interpret the mathematical concept of gradient flow as a continuous-time version of policy iteration.

2. We show that the choice of a Wasserstein-2 trust region in such a setting leads to solving the Fokker-Planck equation in (infinite dimensional) policy space, leading to the concept of *policy transport*. This shows optimal policies are arrived at via diffusion and advection. This also justifies empirical practices such as adding Gaussian noise to gradients.

# 2 GRADIENT FLOWS

## 2.1 ENTROPY-REGULARISED REWARDS

Let $r(a)$ be the reward obtained by taking action $a$. The expected reward with respect to a policy $\pi$ is:

$$K_r(\pi) = \mathbb{E}_\pi\big[r(a)\big] = \int_{\mathbb{A}} r(a)d\pi(a) \tag{1}$$

Shannon entropy is often added as a regularization term to improve exploration and avoid early convergence to suboptimal policies. This gives us the entropy-regularised reward, which is a *free energy* functional, named by analogy with a similar quantity in statistical mechanics [1] :

$$\begin{aligned} J(\pi) &= \int_{\mathbb{A}} r(a)d\pi(a) - \beta \int_{\mathbb{A}} \log \pi(a)d\pi(a) \\ &= K_r(\pi) - \beta H(\pi) \end{aligned} \tag{2}$$

## 2.2 POLICY ITERATION AS GRADIENT FLOW

We are interested in the process of policy iteration, that is, finding a sequence of policies $(\pi_n)$ converging towards the optimal policy $\pi^*$. In this section we follow closely the exposition by Santambrogio. (2015). Policy iteration is often implemented using gradient ascent according to

$$\pi_{k+1} = \pi_k + \tau \nabla J(\pi_k) \tag{3}$$

Rearranging gives the explicit Euler method

$$\frac{\pi_{k+1} - \pi_k}{\tau} - \nabla J(\pi_k) = 0 \tag{4}$$

In this article we are more interested in the implicit Euler method which simply replaces $\nabla J(\pi_k^\tau)$ with $\nabla J(\pi_{k+1}^\tau)$

$$\frac{\pi_{k+1} - \pi_k}{\tau} - \nabla J(\pi_{k+1}) = 0 \tag{5}$$

This is a policy iteration method. If integrated and interpreted as an $L^2$ regularized iterative problem, it is strictly equivalent to finding a solution to the proximal problem:

$$\pi_{k+1} = \arg\min_\pi \frac{||\pi - \pi_k||^2}{2\tau} - J(\pi) \tag{6}$$

Rather than just the $L^2$ distance between policies for constraining and regularization, one can envision the more general case of any policy distance $d$:

$$\pi_{k+1} = \arg\min_\pi \frac{d^2(\pi, \pi_k)}{2\tau} - J(\pi) \tag{7}$$

## 2.3 WASSERSTEIN PROXIMAL MAPPING: POLICY TRANSPORT

$d$ can be chosen as any general metric distance or divergence. For instance, should we choose $d = \sqrt{D_{KL}}$, the square root of the Kullback-Leibler divergence, in the proximal mapping above,

---

[1]In this article we follow the convention of convex analysis and optimal transport, that is, entropy $H$ is taken to be convex, rather than that of information theory with H preceded by a negative sign and concave.

we'd get a policy iteration procedure close in spirit to Schulman's trust region policy optimization Schulman et al. (2015a.).

The case of interest in this paper is when $d = W_2$ is the second Wasserstein distance, that gives the optimal transport cost for cost function $c(x, y) = \frac{1}{2}|x - y|^2$. We therefore do iterative minimization in the *Wasserstein-2* space $\mathbb{W}_2$:

$$
\begin{aligned}
\pi_{k+1} &= \arg\min_\pi \frac{W_2^2(\pi, \pi_k)}{2\tau} - J(\pi) \\
&= \arg\min_\pi \frac{W_2^2(\pi, \pi_k)}{2\tau} - \int_{\mathbb{A}} r(s, a)d\pi + \beta \int_{\mathbb{A}} \log \pi(a|s)d\pi
\end{aligned}
\tag{8}
$$

A gradient flow is obtained in the small step limit $\tau \to 0$. This proximal mapping is an example of Moreau envelope, and remains in a convex optimization setting when $J$ is convex. The Wasserstein distance $W_2$ is defined for pairs of measures $(\mu, \nu)$ seen as marginals of a coupling $\gamma$ by:

$$
\forall (\mu, \nu) \in \mathbb{P}^2, W_2^2(\mu, \nu) = \inf_{\gamma \in \Gamma(\mu,\nu)} \iint |x - y|^2 d\gamma(x, y) = \inf_{X \sim \mu, Y \sim \nu} \mathbb{E}|X - Y|^2
\tag{9}
$$

The optimal coupling $\gamma^*$ in the infimum above is also called the optimal transport plan.

Also note that later, we will reformulate this Monge-Kantorovich problem Kantorovich. (1942) as an equivalent linear problem of inner product minimization in $L^2(\mathbb{P}^2)$:

$$
W_2^2(\mu, \nu) = \inf_{\gamma \in \Gamma(\mu,\nu)} \langle c(x, y), d\gamma(x, y) \rangle
\tag{10}
$$

## 3 DERIVING THE FOKKER-PLANCK EQUATION

Jordan, Kinderlehrer and Otto showed in a seminal result that several partial differential equations can be interpreted as steepest descent, or gradient flows, of functionals in Wasserstein space $\mathbb{W}_2$ (Jordan et al., 1998). Here we compute the PDE associated with the entropy-regularized rewards functional $J$ and its steepest descent within $\mathbb{W}_2$.

### 3.1 THE EULER CONTINUITY EQUATION

If we take the limit of small steps size $\tau$ in the implicit Euler method of equation 5 we get the Cauchy problem

$$
\frac{\partial \pi}{\partial t} = \pi'(t) = \nabla J(\pi(t))
\tag{11}
$$

$$
\pi(0) = \pi_0
\tag{12}
$$

which describes a *gradient flow*.

Just like gradient flows are the continuous-time analogue of discrete gradient descent steps, the Euler continuity equation is the continuous-time analogue of the discrete Euler methods seen earlier.

A classic result in Wasserstein space analysis is that because optimal transport acts on probability measures, it must satisfy conservation of mass. Hence all absolutely continuous curves, or flows of measures, $(\pi_t)$ in $\mathbb{W}_2(\mathbb{P})$ are solutions of the Euler continuity equation. The Euler continuity equation can be seen as the formal continuous-time limit of the Euler implicit method described above

$$
\partial_t \pi_t = -\nabla \cdot (v_t \pi_t)
\tag{13}
$$

where $v_t$ is a suitable vector velocity field ($\nabla \cdot$ is the divergence operator). In the case we are looking at:

$$
v_t = \nabla \left( \frac{\delta J}{\delta \pi}(\pi) \right)(a)
\tag{14}
$$

where $\frac{\delta J}{\delta \pi}(\pi)$ is the *first variation density* defined via the Gateaux derivative as

$$
\int \frac{\delta J}{\delta \pi}(\pi)d\xi = \frac{d}{d\epsilon} J(\pi + \epsilon\xi) \Big|_{\epsilon=0}
\tag{15}
$$

for every perturbation $\xi = \pi' - \pi$.

Substituting $v_t$ into 13 gives us a partial differential equation necessarily of the form:

$$\partial_t \pi = -\nabla \cdot (\pi \nabla \left( \frac{\delta J}{\delta \pi}(\pi) \right)) \tag{16}$$

This is proven rigorously in Jordan et al. (1998).

## 3.2 THE FOKKER-PLANCK EQUATION

It remains to compute the first variation density $\frac{\delta J}{\delta \pi}(\pi)$ for entropy regularised policy gradients.

First, the linear part $K_r$, or *potential energy* has first variation given naturally by

$$K_r(\pi) = \int_{\mathbb{A}} r d\pi \Rightarrow \frac{\delta K_r}{\delta \pi}(\pi) = r \tag{17}$$

Second, the entropy part $H$ is a special case $H = U_{t \log t}$ of the general *internal energy* density functional:

$$U_f(\pi) = \int_{\mathbb{A}} f \left( \frac{d\pi(a)}{da} \right) da \Rightarrow \frac{\delta U_f}{\delta \pi}(\pi) = f'(\pi) \tag{18}$$

In the case of entropy, $f(t) = t \log t$, $f'(t) = 1 + \log t$, and therefore

$$\frac{\delta H}{\delta \pi}(\pi) = 1 + \log \pi. \tag{19}$$

Finally we require the gradient of this first variation density, given by:

$$\nabla \left( \frac{\delta J}{\delta \pi} \right) = \nabla r - \beta \nabla (1 + \log \pi) = \nabla r - \beta \frac{\nabla \pi}{\pi} \tag{20}$$

The gradients $\delta / \delta \pi$ are functional, whereas the gradients $\nabla$ are action-gradients $\nabla_a$ with respect to actions $a \in \mathbb{A}$.

Plugging this into 16 gives us the partial differential equation associated with steepest descent within $W_2$ for entropy-regularized rewards:

$$\partial_t \pi = -\nabla \cdot (\pi \nabla r) + \beta \Delta \pi \tag{21}$$

## 4 INTERPRETATION

### 4.1 CONVERGING TO THE OPTIMAL POLICY

The entropy-regularized rewards $J$ is convex in the policy $\pi$, which means there is a single optimal policy. The optimal policy will be achieved as long as each step improves the policy and this will be the case as long as the steps taken as are small enough.

Given that we converge to the optimal policy, we know that at optimality $\partial_t \pi = 0$. Using equation 16 then gives us a necessary condition for the optimal policy $\pi^*$:

$$-\nabla \cdot (\pi \nabla \left( \frac{\delta J}{\delta \pi}(\pi) \right)) = 0 \tag{22}$$

By setting $\frac{\delta J}{\delta \pi}(\pi) = 0$ and substituting in 20 we get

$$\nabla r = \beta \frac{\nabla \pi^*}{\pi^*} = \beta \nabla \log \pi^* \tag{23}$$

which gives us the optimal policy

$$\pi^* \propto e^{r/\beta} \tag{24}$$

This is the Gibbs measure of the rewards density per action - also seen as an *energy-based* policy, in line with the static result of Sabes & Jordan. (1996). The unregularized case $\beta = 0$ appears degenerate.

## 4.2 THE FOKKER-PLANCK EQUATION

The gradient flow associated with the pure entropy functional $\beta H$ is the heat equation $\partial_t \pi = \beta \Delta \pi$. Here the Laplacian $\Delta$ is in action space. This is one of the key messages of the derivations we have done in this part: *the Wasserstein gradient flow turns the entropy into the Laplacian operator*[2].

For the full, entropy-regularized reward $J(\pi)$, there is an extra transport term generated by the rewards, and the PDE is therefore the Fokker-Planck equation. This means that taking policy gradient ascent steps in $\mathbb{W}_2$ (according to equation 8), is equivalent to solving the Fokker-Planck equation for the policy $\pi$ with potential field equal to the gradient of rewards $r$. Alternately, we can say, the policy undergoes diffusion and advection - it concentrates around actions with high reward.

## 4.3 BROWNIAN MOTION AND NOISY GRADIENTS

A partial differential equation for diffuse measures also admits a particle interpretation. The result above can also be written, through Ito's lemma Revuz & Yor. (1999), as the stochastic diffusion version of the Fokker-Planck equation Jordan et al. (1998):

$$d\Pi_t = -\nabla r(\Pi_t) dt + \sqrt{2\beta} dB_t \tag{25}$$

with $\Pi_t$ a finite dimensional discretization of policy $\pi_t$, and $B_t$ a Brownian motion of same dimensionality. In that case, the density of solutions verifies equation 21 - formally, one replaces increments of the Brownian motion $dB_t$ by its *infinitesimal generator*, the Laplacian $\frac{1}{2}\Delta$.

This stochastic differential equation can also be seen as a Cauchy problem of on-policy rewards maximization, this time with added isotropic Gaussian policy noise $\sqrt{2\beta} dB_t$.

Discretizing again - but in the time variable rather than the action space - one writes, with an explicit method:

$$\Pi_{n+1} - \Pi_n = -\nabla r(\Pi_n) + \sqrt{2\beta} \cdot \mathbf{N}(0, \mathbf{I}_q) \tag{26}$$

This is just noisy stochastic *action-gradients* ascent on the rewards field. This shows a link between entropy-regularization and noisy gradients. It also suggests a possible technique for implementation. The key issue to overcome is to generate gradient noise in parameter space that is equivalent to isotropic Gaussian policy noise.

## 4.4 RELATIONSHIP BETWEEN WASSERSTEIN AND KULLBACK-LEIBLER DISTANCES

We note the Kullback-Leibler and Wasserstein-2 problems are related by the *Talagrand p-inequalities* Gozlan & Léonard. (2010), which for $p = 2$ and under some conditions, ensure that for some constant $C$ and given a suitable reference measure $\nu$, $\forall \mu \in \mathbb{P}(X), W_2^2(\mu, \nu) \leq C \cdot D_{KL}(\mu, \nu)$. This justifies the square root in $d = \sqrt{D_{KL}}$ in the proximal mappings studied earlier (equation 8), but more would be beyond the scope of this article.

## 4.5 ENERGY-BASED POLICIES AS GIBBS MEASURES

Since the first variation process $\frac{\delta J(\pi)}{\delta \pi}$ is known, and we derive it explicitly earlier, we can use a variational argument specific to $\mathbb{W}_2$ (and invalid in $\mathbb{W}_1$ !). We admit (Santambrogio. (2015)) that the solution of the minimization problem has to satisfy:

$$-\frac{\delta J}{\delta \pi}(\pi) + \frac{\phi_{\mathbb{W}_2}}{\tau} = constant \tag{27}$$

with $\phi_{\mathbb{W}_2}$ a Kantorovich potential function for transport cost $c(x, y) = \frac{1}{2}|x - y|^2$. One useful way to think of the Kantorovich potential is that it is a function, whose gradient field generates the difference between the optimal transport plan $T$ and the identity, according to the equation $T(x) = x - \nabla\phi(x)$.

It is well known Revuz & Yor. (1999) that the Gibbs distribution is the invariant measure of the stochastic differential equation above. Therefore we expect it to play a role of primary importance.

---

[2]This is part of what's known as *Otto calculus*.

In fact, the solution of the $\mathbb{W}_2$ gradient flow with discrete steps is known explicitly Santambrogio. (2015) if we know the successive Kantorovich transport potentials associated:

$$\pi_n(a|s) \sim e^{\frac{r(s,a)+\phi_n(a)}{\beta}} \underset{n\to\infty}{\to} e^{\frac{r(s,a)}{\beta}} \sim \pi^*(a|s) \tag{28}$$

In practice, deriving the $\mathbb{W}_2$ optimal transport as well as its cost at each gradient flow step is numerically instable and computationally expensive. Furthermore, numerical estimators for the gradients of Wasserstein distances have been found to be biased; alternatives such as the Cramer distance behave better in practice but not in theory Bellemare et al. (2017b). (To our knowledge, the gradient flow of the Cramer distance is not known, and no results exist that relate it to the entropy and Fisher information functionals). In appendix, we use fast approximate algorithms in their small parameter regime. We show that another Gibbs measure, the two-dimensional *kernel* $e^{-c/\epsilon}$, where $c$ is the ground transport cost, and $\epsilon$ an auxiliary regularization strength, arises naturally in this context, and leads to taking successive Kullback-Leibler steepest descent steps in the coupling, in a spirit close to trust region policy optimization, but using optimal transport and the *Sinkhorn* algorithm.

## 5 RELATED WORK

Optimal transport, and the study of Wasserstein distances, is a very active research field. Foundational developments are found in Villani's reference opus Villani. (2008). The gradient flows perspective is presented in Ambrosio's book Ambrosio et al. (2006) for a complete theoretical treatment, and in Santambrogio Santambrogio. (2015) for a more applied view including a presentation of the Jordan-Kinderlehrer-Otto result. A classic reference for connecting Brownian motions and partial differential equations is Revuz-Yor Revuz & Yor. (1999). Efficient algorithms for regularized optimal transport were first explored by Cuturi Cuturi. (2013), and then Peyré Peyré (2015) who showed the equivalence to steepest descent of KL with respect to the smoothed Gibbs ground cost, and its formulation as a convex problem. Carlier Carlier et al. (2015) gives proofs of $\Gamma$-convergence of $W_{2,\epsilon}$ to $W_2$. Léonard Léonard. (2014) makes the connection with the Schrodinger problem Schrodinger. (1931) and concentration of measure.

In the context of neural networks, partial differential equations and convex analysis methods are covered by Chaudhari Chaudhari et al. (2017). The Monge-Kantorovich duality in the $W_1$ case, and Wasserstein representation gradients, are applied to generative adversarial networks by Arjovsky Arjovsky et al. (2017). The $W_2$ connection with generative models is studied by Bousquet Bousquet et al. (2017). Similarly, Genevay et al Genevay et al. (2017) define Minimum Kantorovich Estimators in order to formulate a wide array of machine learning problems in a Wasserstein framework.

## 6 DISCUSSION AND FURTHER WORK

We have used tools of quadratic optimal transport in order to provide a theoretical framework for entropy-regularized reinforcement learning, under the strongly restrictive assumption of maximising one-step returns. There, we equate policy gradient ascent in Wasserstein trust regions with the heat equation using the JKO result. We show advection and diffusion of policies towards the optimal policy. This optimal policy is the Gibbs measure of rewards, and is also the stationary distribution of the heat PDE. Recast as a stochastic Brownian diffusion, this helps explain recent methods used empirically by practitioners - in particular it sheds some light on the success of noisy gradient methods. It also provides a speculative mechanism besides the central limit theorem for why Gaussian distributions seem to arise in practice in distributional reinforcement learning Bellemare et al. (2017a).

Our contribution largely consists in highlighting the connection between the functional of reinforcement learning and these mathematical methods inspired by statistical thermodynamics, in particular the Jordan-Kinderlehrer-Otto result. While we have aimed to keep proofs in this paper as simple and intuitive as possible, an extension to the n-step returns (multi-step) case is the most urgent and obvious line of further research. Finally, exploring efficient numerical methods for heat equation flows compatible with function approximation, are directions that will also be considered in future research.

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

## A  JUSTIFYING STEEPEST DESCENT OF RELATIVE ENTROPY : ENTROPIC REGULARIZATION OF $W_2$ ITSELF

Here we show that while taking discrete proximal steps according to distance $d = W_2$ is ill-advised, we can perform optimal transport with respect to the *entropic regularization* of the Wasserstein distance itself. This is a second layer of entropic regularization.

If we let $\epsilon$ be a small positive real number, then we define $W_{2,\epsilon}^2$ as a regularization of the $W_2$ minimization in equation 9:

$$\forall (\mu, \nu) \in \mathbb{P}^2, W_{2,\epsilon}^2(\mu, \nu) = \inf_{\gamma \in \Gamma(\mu, \nu)} \iint |x - y|^2 d\gamma(x, y) - \epsilon \overline{H}(\gamma) \tag{29}$$

with the entropy $\overline{H}(\gamma)$ now extended to two-dimensional coupling space $\Gamma(\mu, \nu)$ as, in the discretized case, with $card\mathbb{A} = q$,

$$\overline{H}(\gamma) = - \sum_{i,j=1}^{q} \gamma_{i,j} \Big( \log(\gamma_{i,j}) - 1 \Big) \tag{30}$$

and by an analogue of continuity named $\Gamma$-convergence Carlier et al. (2015), $W_{2,\epsilon}^2(\mu, \nu) \underset{\epsilon \to 0}{\to} W_2^2(\mu, \nu)$. In fact, $W_{2,\epsilon}^2$ converges exponentially Cominetti & Martin. (1994) fast towards $W_2^2$ as $\epsilon \to 0$. $W_{2,\epsilon}^2$ is not a distance anymore, but rather a divergence. Yet it enjoys better numerical properties, and enables closed-form solutions for various problems. We do not have a linear Monge-Kantorovich minimization program anymore, but rather a strictly convex program. With $c(x, y) = \frac{1}{2}|x - y|^2$:

$$\forall (\mu, \nu) \in \mathbb{P}^2, W_{2,\epsilon}^2(\mu, \nu) = \inf_{\gamma \in \Gamma(\mu, \nu)} \Big( \langle c, \gamma \rangle - \epsilon H(\gamma) \Big) \tag{31}$$

This change from a linear to a convex problem makes the solution set better conditioned numerically; the solution does not have to lie on a vertex of a convex polytope (by analogy with the simplex algorithm, see Nesterov et al. (1994)) anymore, and therefore, is more robust to initial conditions. In practice, $\epsilon$ cannot be chosen too small or these stability properties are lost. A certain amount of smoothing is to be tolerated, which is acceptable in the reinforcement learning context, due to the inherent uncertainty on the rewards distribution. Returning to our optimization problem, moving the inner product bracket inside the $H$ part turns the expression into a single KL divergence. This yields the equivalent problem, as detailed in Peyré (2015)

$$\forall (\mu, \nu) \in \mathbb{P}^2, W_{2,\epsilon}^2(\mu, \nu) = \inf_{\gamma \in \Gamma(\mu, \nu)} H_{e^{-c/\epsilon}}(\gamma) \tag{32}$$

At this stage, the link with earlier sections becomes intuitively very clear, since the reference measure $e^{-c/\epsilon}$ is for $c_{W_2}(x, y) = \frac{1}{2}|x - y|^2$ simply the *heat kernel* $e^{\frac{-|x-y|^2}{2\epsilon}}$. It is therefore not surprising the evolution gradient flows considered earlier were linked to the heat equation. Performing JKO stepping from from $d^2 = W_{2,\epsilon}^2$ rather than $W_2^2$ reads

$$\pi_{k+1}^{\tau,\epsilon} = \arg\min_{\pi} \frac{W_{2,\epsilon}^2(\pi, \pi_k^{\tau,\epsilon})}{2\tau} + F(\pi) \tag{33}$$

instead of equation 8. Combining both definitions therefore gives the problem

$$\gamma^* = \arg\min_{\gamma \in \Gamma(\mu, \nu)} H_{e^{-c/\epsilon}}(\gamma) + \frac{2\tau}{\epsilon} F(\pi \cdot \mathbf{1}) \tag{34}$$

to be solved in 2-d coupling space. With this entropic smoothing, we can now re-cast the optimal transport problem as a Kullback-Leibler problem, trading a single optimal transport proximal step for several, 'fast' KL steps. This is done next section using iterative convex projection algorithms.

## B    DERIVATION OF THE SINKHORN ALGORITHM

### B.0.1    THE BREGMAN ALGORITHM

This algorithm is used in convex optimization for iterative projections. The method generalizes the computation of the projection on the intersection of convex sets. Assume we give ourselves a convex function $\Psi$ and that we consider the associated Bregman divergence $D_\Psi$ Amari. (2016) defined by

$$\forall(\pi, \xi) \in \mathbb{P}^2, \quad D_\Psi(\pi|\xi) = \Psi(\pi) - \Psi(\xi) - \langle\nabla\Psi(\xi), \pi - \xi\rangle \tag{35}$$

We look to minimize this Bregman divergence $D_\Psi$ on the intersection of convex sets $C = \cap \ \ C_i$. In our case of interest there are two such sets $C_1$ and $C_2$. For $y$ a given point, or function, we solve for

$$\inf_{x \in C} D_\Psi(x, y) \tag{36}$$

or equivalently with $\phi_1$ and $\phi_2(\pi)$ playing the role of indicator barrier functions

$$\inf_{\pi \in \mathbb{P}} \ \ D_\Psi(\pi|\xi) + \phi_1(\pi) + \phi_2(\pi) \tag{37}$$

In the case where $\Psi = H$, we get $\nabla\Psi = \log$, $\nabla\Psi^* = \exp$ through Legendre transform gradient bijection, and $D_\Psi = H_\Psi = D_{KL}(\cdot|\Psi)$.

The Bregman algorithm Bregman. (1967) simply consists in solving the problem 36 by iteratively performing projection on each of the sets $C_i$ in a cyclical manner, therefore building the sequence

$$x_{n+1} = P_{C_{[n]}}^{D_\Psi}\left(x_n\right) \underset{n\to\infty}{\to} x^* = \inf_{x \in C} D_\Psi(x, y) \tag{38}$$

with $[n]$ the modulo operator ensuring cyclicality of the projections. Therefore, any problem that can be cast under the convex Bregman form 36 can be solved by taking many steepest descent steps. We now proceed to explicit the $P_{C_{[n]}}^{D_\Psi}$ operators, which in our case are $P_{C_{[n]}}^{KL}$ KL proximal steps, and integrate them into an efficient practical algorithm.

### B.0.2    THE SINKHORN ALGORITHM

We start with the need to minimize the convex form $\frac{1}{\epsilon}\sum_{ij} p_{ij}\log p_{ij} + p_{ij}c_{ij}$ , subject to marginal constraints that the discretized measure $\mu$ is transported by $p$ onto $\nu$. From a matrix perspective this translates into the two following constraints: that the sum in column of $P$ being equal to vector $\mu$, and the sum across lines of $P$ equal to $\nu$:

$$P \in U(\mu, \nu) = \{M \in \mathbb{R}^{q \times q}, \quad M\mathbf{1}_q = \mu, \quad M^T\mathbf{1}_q = \nu\} \tag{39}$$

The matrix set $U(\mu, \nu)$ is convex. We can form the Lagrangian of this optimization problem in $p_{ij}$ using vector Lagrangian multipliers $\alpha, \beta$,

$$\mathcal{L}(P, \alpha, \beta) = \frac{1}{\epsilon}\sum_{ij} p_{ij}\log p_{ij} + p_{ij}c_{ij} + \alpha^T(P\mathbf{1}_q - \mu) + \beta^T(P^T\mathbf{1}_q - \nu) \tag{40}$$

A necessary condition for optimality is then

$$\forall(i, j), \quad \frac{\partial\mathcal{L}}{\partial p_{ij}^\epsilon} = 0 \quad \Rightarrow \quad p_{ij}^* = \exp\left(-\frac{1}{2} - \frac{\alpha_i}{\epsilon}\right)\exp\left(-\frac{c_{ij}}{\epsilon}\right)\exp\left(-\frac{1}{2} - \frac{\beta_j}{\epsilon}\right) \tag{41}$$

Hence we have shown that the optimal coupling is a diagonal scaling of the ground cost's Gibbs kernel. With the two positive vectors $u, v$ defined as $diag(u) = e^{-\frac{1}{2} - \frac{\alpha_i}{\epsilon}}$ and $diag(v) = e^{-\frac{1}{2} - \frac{\beta_j}{\epsilon}}$, this is formally $p_{ij}^* = u_i\exp(-c_{ij}/\epsilon)v_j$. Once this is derived, the iterative convex projections framework of the Bregman algorithm enables us to derive the full Sinkhorn method.

If one recasts the entropic transport problem

$$\min_{\gamma \in \Gamma}\left(\sum_{i,j=1}^N c_{i,j}\gamma_{i,j} + \epsilon H(\gamma|(\mu_i\nu_j)_{ij})\right) \tag{42}$$

as

$$\min_{\gamma \in \Gamma} \quad H(\gamma | \bar{\gamma}) \quad \Rightarrow \quad \gamma_{i,j}^{*,\epsilon} = \exp\left(-c_{i,j}/\epsilon\right)\mu_i \nu_j \tag{43}$$

then this reads as a two-dimensional KL projection algorithm of point $\bar{\gamma}$ on set of marginal constraints $C_{1,\mu}$ enforcing $u \odot (Kv) = \mu$, and $C_{2,\nu}$ enforcing $v \odot (K^T u) = \nu$. Therefore

$$\gamma_{i,j}^{*,\epsilon} = u_i \bar{\gamma}_{i,j} v_j = u_i \cdot \mu_i \exp\left(-c_{i,j}/\epsilon\right)\nu_j \cdot v_j \tag{44}$$

and ultimately yields the Sinkhorn balancing algorithm:

$$u \leftarrow \left(\frac{\mu_i}{\sum_j \bar{\gamma}_{i,j} v_j}\right)_i = \frac{\mu}{Kv} \qquad v \leftarrow \left(\frac{\nu_j}{\sum_i \bar{\gamma}_{i,j} u_i}\right)_j = \frac{\nu}{K^T u} \tag{45}$$

these two updates being merged in Algorithm 1.

### B.1   THE SINKHORN ALGORITHM

The Sinkhorn algorithm is a fast, iterative algorithm for optimal transport; it mostly involves matrix multiplications and vector operations such as term-by-term division, and as such scales extremely well on GPU platforms. This makes it possible to use the Sinkhorn algorithm to numerically approximate optimal couplings. We give its outline below, the interested reader can find more details can be found in Cuturi's original article Cuturi. (2013); Sinkhorn & Knopp. (1967), or in Frogner's version applied to deep learning Frogner et al. (2015).

We will want to compute the optimal coupling, transport cost, and gradient pertaining to distance $W_{2,\epsilon}^2$. First we remember the regularized transport problem as per equation 31. The 2-dimensional, relaxed coupling $\gamma$ can be discretized to a 2-dimensional matrix $P_\epsilon$ with entries $(p_{i,j})$. We show (see Appendix) that necessarily

$$P^* = diag(u)\mathbf{K}diag(v) \quad \mathbf{K} = e^{-\frac{C}{\epsilon}} \tag{46}$$

where the matrix exponential of the ground cost is taken term-by-term. Recalling the equality constraints on the row and column sums given by $\mu$ and $\nu$ in 39, we find that we have to solve a *matrix balancing* problem, using the terminology of linear algebra. Once we have formed $\mathbf{K}$, and have policies $\mu$ and $\nu$ as inputs, we can run through iterations of the Sinkhorn algorithm till convergence to a fixed point. This is done below and runs a one-line *while* loop on vector $(x)_i$, the component-wise inverse of $(u)_i$. Dotted operations are taken component-wise:

---

**Algorithm 1** Computation of policy transport $W_c^\epsilon(\mu, \nu)$ by Sinkhorn iteration.

> **Input** C, $\epsilon$, $\mu$, $\nu$.
> $I = (\mu > 0)$; $\mu = \mu(I)$; $c = c(I, :)$ ; K=exp(-$C/\epsilon$)
> x=ones(length($\mu$),size($\nu$,2))/length($\mu$);
> **while** x has not converged **do**
>     x=diag(1./$\mu$)*K*($\nu$.*(1./(K'*(1./x))))
> **end while**
> u=1./x; v=$\nu$.*(1./(K'*u))
> $W_c^\epsilon(\mu,\nu)$=sum(u.*((K.*C)*v))
> $\frac{\partial W_c^\epsilon(\mu,\nu)}{\partial \mu} = \epsilon \log u$ (up to a constant parallel shift)

---

The Sinkhorn algorithm converges linearly. Its theoretical justification is that it can be seen as an instance of the iterative convex projections explained previous section. It is critical to notice that the distance $W_{2,\epsilon}$ is differentiable in the policy, unlike $W_2$. The vector $u$ above is not unique; but one suitable gradient of the Wasserstein distance with respect to the first variable policy is known, and given by the formula in the algorithm above, simply proportional to the element-wise log of scaling vector $u$. This closed form differentiation allows us to perform gradient descent in Sinkhorn layers embedded in neural network systems. In general, this gradient, just like vector $u$, is defined up to a constant shift only; the normalizing shift generally found in the literature is

$$\frac{\partial W_c^\epsilon(\mu,\nu)}{\partial \mu} = \epsilon \log u - \epsilon \frac{\log u^T \mathbf{1}}{\mathbf{K}} \mathbf{1} \tag{47}$$

that makes $u$ tangent to the simplex Frogner et al. (2015). Under this form, the algorithm is compatible with function approximation, where policy $\mu$ is a function of a parameter and reads $\mu_\theta$. We note that another possibility to create this compatibility would be to unroll a fixed number of iterations of the algorithm, as they are effectively matrix and vector operations, as has already been done with generative adversarial networks and deep Q-networks. We hypothesize that learning with a Wasserstein loss, in a continuous action state setting, will help agents pick actions that are semantically close to the optimum action, therefore increasing policy quality, and reducing the 'unnaturalness' of policy mistakes. It is our hope that a Wasserstein loss, by implying relevant semantic directions in action space, will speed up convergence and training of reinforcement learning agents. In practice, we are still limited by our fundamental assumption that the MDP and the statewise rewards density $a \rightarrow r(s, a)$ are known. Possibilities such as bootstrapping the rewards density distribution exist, and will be explored practically in further work.

