# OpenReview forum: "Diffusing Policies : Towards Wasserstein Policy Gradient Flows"
_ICLR.cc/2018/Conference — Reject_

### Official Review · AnonReviewer1 · 2017-11-26
**Interesting insights on policy gradient flows but the novel contributions are unclear**

**Rating:** 4
**Confidence:** 3

**Review:**

The paper ‘Diffusing policies: Towards Wasserstein policy gradient flows’ explores
the connections between reinforcement learning and the theory of quadratic optimal transport (i.e.
using the Wasserstein_2 as a regularizer of an iterative problem that converges toward
an optimal policy). Following a classical result from Jordan-Kinderlehrer-Otto, they show that
the policy dynamics are governed by the heat equation, that translates in an advection-diffusion
scheme. This allows to draw insights on the convergence of empirical practices in the field.

The paper is clear and well-written, and provides a comprehensive survey of known results in the
field of Optimal Transport. The insights on why empirical strategies such as additive gradient noise
are very interesting and helps in understanding why they work in practical settings. That being said,
most of the results presented in the paper are already known (e.g. from the book of Samtambrogio or the work
of G. Peyré on entropic Wasserstein gradient flows) and it is not exactly clear what are the original
contributions of the paper. The fact that the objective is to learn policies
has little to no impact on the derivations of calculus. It clearly suggests that the entropy
regularized Wasserstein_2 distance should be used in numerical experiments but this point is not
supported by experimental results. Their direct applications is rapidly ruled out by highlighting the
computational complexity of solving such gradient flows but in the light of recent papers (see
the work of Genevay https://arxiv.org/abs/1706.00292 or another paper submitted to ICLR on large scale optimal transport
https://openreview.net/forum?id=B1zlp1bRW) numerical applications should be tractable. For these reasons
I feel that the paper would clearly be more interesting for the practitioners (and maybe to some extent
for the audience of ICLR) if numerical applications of the presented theory were discussed or sketched
in classical reinforcement learning settings.

Minor comments:
 - in Equation (10) why is there a ‘d’ in front of the coupling \gamma ?
 - in Section 4.5, please provide references for why numerical estimators of gradient of Wasserstein distances
are biased.

---

> ### Author Response · Authors · 2018-01-05
> **RE : Interesting insights**
>
> We thank the reviewer for their interest and for their comments on clarity and style.
>
> We do agree the paper would benefit from practical results ; we feel there is value from a theoretical standpoint in exposing the connections with proximal mappings and gradient flow PDEs to the RL community, as we hope the general method of equating proximal regularizer, gradient flow PDE, and related stochastic process will become more widespread.
>
> We are also thankful for your referencing of https://arxiv.org/abs/1706.00292 and https://openreview.net/forum?id=B1zlp1bRW, both of which we were unaware of as of time of writing this paper, obviously. We are indeed hopeful to remediate the lack of empirical results due to both tractability of large-scale optimal transport, and of compatibility of function approximation methods with Fokker-Planck diffusion. We will endeavour to include insights from these papers in further work.
>
> Finally, the d_\gamma in equation (10) is a notation artifact made to link with the d_\gamma in equation (9), but it probably is cleaner to correct and omit it. Regarding biased sample gradients of the Wasserstein distance, we do provide our article's fifth reference - a key recent paper that has highlighted this issue is Bellemare et al.'s https://arxiv.org/abs/1705.10743 ; we will clarify that we are referring to sample gradients bias here.

---

### Official Review · AnonReviewer4 · 2017-11-27
**Diffusing Policies : Towards Wasserstein Policy Gradient Flows**

**Rating:** 5
**Confidence:** 3

**Review:**

In this paper the authors studied policy gradient with change of policies limited by a trust region of Wasserstein distance in the multi-armed bandit setting. They show that in the small steps limit, the policy dynamics are governed by the heat equation (Fokker-Planck equation). This theoretical result helps us understand both the convergence property and the probability matching property in policy gradient using concepts in diffusion and advection from the heat equation. To the best of my knowledge, this line of research was dated back to the paper by Jordan et al in 1998, where they showed that the continuous control policy transport follows the Fokker-Planck equation. In general I found this line of research very interesting as it connects the convergence of proximal policy optimization to optimal transport, and I appreciate seeing recent developments on this line of work.

In terms of theoretical contributions, I see that this paper contains some novel ideas in connecting gradient flow with Wasserstein distance regularization to the Fokker-Planck equation. Furthermore its interpretation on the Brownian diffusion processes justifies the link between entropy-regularization and noisy gradients (with isotropic Gaussian noise regularization for exploration). I also think this paper is well-written and mathematically sound. While I understand the knowledge of this paper based on standard knowledge in PDE of diffusion processes and Ito calculus, I am not experienced enough in this field to judge whether these contributions are significant enough for a standalone contribution, as the problem setting is limited to multi-armed bandits.

My major critic to this paper is its practical value. Besides the proposed Sinkhorn-Knopp based algorithm in the Appendix that finds the optimal policy as fixed point of (44), I am unsure how these results lead to more effective policy gradient algorithms (with lower variance in gradient estimators, or with quasi-monotonic performance improvement etc.). There are also no experiments in this paper (for example to compare the standard policy gradient algorithm with the one that solves the Fokker-Planck equation) to demonstrate the effectiveness of the theoretical findings.

---

> ### Author Response · Authors · 2018-01-05
> **RE : practical value**
>
> Thank you very much for your insights and comments, as well as encouraging words on soundness and writing style. We are in agreement that the paper would benefit both from a theoretical standpoint if we could extend the results to the n-step returns setting, and from a practical perspective if we could an exhibit a numerically tractable algorithm using the Wasserstein policy iteration. While theoretical difficulties have arisen in combining neural-network based function approximation with the Fokker-Planck PDE, we do share this reviewer's concern and urgency on that point, and are currently undergoing work on this in a tabular setting.

---

### Official Review · AnonReviewer3 · 2017-11-27
**Important topic but the work is a presentation of known material**

**Rating:** 4
**Confidence:** 4

**Review:**

The main object of the paper is the (entropy regularized) policy updates. Policy iterations are viewed as a gradient flow in the small timestep limit. Using this, (and following Jordan et al. (1998)) the desired PDE (Equation 21) is obtained. The rest of the paper discusses the implications of Equation 21 including but not limited to what happens when the time derivative of the policy is zero, and the link to noisy gradients.

Even though the topic is interesting and would be of interest to the community, the paper mainly presents known results and provides an interpretation from the point of view of policy dynamics. I fail to see the significance nor the novelty in this work (esp. in light of  Jordan et al. (1998) and Peyre (2015)).

That said, I believe that exposing such connections will prove to be useful, and I encourage the authors to push the area forward. In particular, it would be useful to see demonstrations of the idea, and experimental justifications even in the form of references would be a welcome addition to the literature.

---

> ### Author Response · Authors · 2018-01-05
> **Thank you for your review and comments.**
>
> Indeed the calculations of sections 3 are found in the major work of Jordan et al. (1998) ; however, it is to our knowledge the first time that the entropy-regularized policy gradient functional is examined in a Wasserstein trust region context (which explains why no references were given for empirical work) in the reinforcement learning context. We do respectfully agree with the reviewer that adding empirical results is the most urgent line of further work.
>
> We do state clearly that 'Our contribution largely consists in highlighting the connection between the functional of reinforcement learning and these mathematical methods inspired by statistical thermodynamics, in particular
> the Jordan-Kinderlehrer-Otto result.' in the discussion. However, and as was stated by another reviewer ('Furthermore its interpretation on the Brownian diffusion processes justifies the link between entropy-regularization and noisy gradients (with isotropic Gaussian noise regularization for exploration)', we believe that the SDE interpretation is new and gives theoretical and intuitive grounding to such articles as https://arxiv.org/abs/1706.10295 and https://arxiv.org/pdf/1707.06887.pdf. Similarly the diffusive nature of convergence to the energy-based policies of Sabes and Jordan was not previously known to us; and we hope the method we have used opens up several new possibilities of continuous relaxations of trust-region RL settings via SDEs and PDEs.

---

### Decision · Program_Chairs · 2018-01-29
**ICLR 2018 Conference Acceptance Decision**

**Decision:**

Reject

**Comment:**

Dear authors,

The authors all agreed that this was an interesting topic but that the novelty, either theoretical or empirical, was lacking. This, the paper cannot be accepted to ICLR in its current state but I encourage the authors to make the recommended updates and to push their idea further.